# Post-Exercise Whey Protein Supplementation: Effects on IGF-1, Strength, and Body Composition in Pre-Menopausal Women, a Randomised Controlled Trial

**DOI:** 10.3390/nu17122033

**Published:** 2025-06-18

**Authors:** Marc Murray, Lara Vlietstra, Alyssa M. D. Best, Stacy T. Sims, James A. Loehr, Nancy J. Rehrer

**Affiliations:** 1School of Physical Education Sport & Exercise Sciences, University of Otago, Dunedin 9054, New Zealand; murma055@student.otago.ac.nz (M.M.); lara.vlietstra@otago.ac.nz (L.V.); alyssabest06@gmail.com (A.M.D.B.); 2Sport Performance Research Institute New Zealand, Auckland University of Technology, Auckland 1010, New Zealand; stacy.sims@gmail.com; 3Stanford Lifestyle Medicine, Stanford Prevention Research Center, Department of Medicine, Stanford University School of Medicine, Stanford, CA 94305, USA; 4Department of Physiology, School of Biomedical Sciences, University of Otago, P.O. Box 56, Dunedin 9054, New Zealand; jim.loehr@otago.ac.nz

**Keywords:** exercise, protein, supplementation, IGF-1, women, pre-menopausal

## Abstract

**Introduction:** The purpose of this study was to evaluate effects of post-exercise protein supplementation with combined resistance and interval training on total insulin-like growth factor-1 (IGF-1) concentration, strength (3RM), and body composition (DXA) in untrained pre-menopausal women. **Methods**: Twenty-seven women (33.6 ± 9.2 years, 69.4 ± 12.4 kg, 25.5 ± 3.7 kg/m^2^) were randomised into a control (CON) or moderate protein group (PRO) (3 g, 24 g, resp.) and completed twelve weeks of upper-body resistance (2×/week) and high-intensity interval cycle training (3×/week). Linear mixed-effects model analyses were conducted. **Results**: PRO had a greater daily protein intake (5.0 ± 16.6 g, 20.5 ± 13.9 g, CON, PRO, resp., *p* = 0.025), with no change in IGF-1 (−6.0 ± 27.7 µg/L, −2.1 ± 27.8 µg/L, CON, PRO, resp., *p* = 0.920). Total lean mass increased (0.84 ± 0.80 kg, 0.56 ± 1.4 kg, CON, PRO, resp., *p* = 0.009), and all strength measures increased in both groups (19–113%, *p* < 0.05). **Conclusions**: Untrained women can increase strength and lean mass over twelve weeks of combined resistance and interval training. Post-exercise protein supplementation had little effect, despite increasing protein intake by ~20 g/day in the PRO group. IGF-1 was not associated with any outcome measure.

## 1. Introduction

Post-exercise protein intake has been increasingly popular, and although research into its effects has also increased, controlled trials amongst healthy, pre-menopausal adult women are limited [1,2]. The timing and composition of nutritional supplementation can both be important for adaptations to exercise training. Protein consumption is not only vital for muscle growth [3,4] but can also influence hormones and synthetic pathways to elicit and maximise adaptations within the body, especially when combined with physical exercise [5]. While resistance exercise stimulates muscle protein synthesis [6], the addition of post-exercise protein consumption further enhances the muscle protein synthetic response through an increased plasma amino acid concentration, which is necessary to repair and promote new tissue growth and maintain a net positive protein balance [7]. The key protein responsible for regulation of these processes is the mammalian target of rapamycin complex 1 (mTORC1), which is an important nutrient sensor and regulates protein synthesis [8], and whey protein appears to induce greater muscle protein synthesis, via enhanced mTOR phosphorylation and downstream signaling, than other amino acid mixtures [9]. In addition to resistance exercise and protein/amino acid composition and concentration, cellular energy status and other growth factors can stimulate mTORC1 activity [10].

One such growth factor is insulin-like growth factor-1 (IGF-1). IGF-1 is a polypeptide hormone with the ability to act as an endocrine (circulating) and paracrine (local acting) hormone [11] that has a multitude of roles within the body. IGF-1 possesses significant anabolic and insulin-sensitising effects, including effects on protein synthesis, which play a role in muscle development and repair [12]. The complete understanding of IGF-1 and its response to exercise is still unclear, although some of the literature has concluded that different exercise types and intensities may influence its response [13]. While resistance exercise has demonstrated acute increases in IGF-1 across a variety of populations [14,15], longitudinal studies are conflicting, primarily due to various factors like sex [16], training status [17,18], age [19], and nutritional status [20]. Some studies have shown baseline concentrations of total IGF-1 to be inversely associated with IGF-1 change in response to various exercise modes [21,22], which is important considering that young women (18–45 years) tend to display higher circulating IGF-1 concentrations compared to age-matched men [23,24]. Previous evidence has demonstrated that younger women can still gain increases in total IGF-1 concentration in response to exercise [25], further adding to the heterogeneity seen within the literature surrounding IGF-1.

Furthermore, aerobic training studies have presented similar results to resistance training, with some evidence indicating increased total IGF-1 concentration [26] and other studies indicating a decrease [22,27] or no change [28] in concentration. Only one of the studies mentioned previously has used a female-only population sample, which was in adolescent females (15–17 years) [27], demonstrating the paucity of data in female exercise science.

Similar to resistance training, the intensity at which aerobic exercise is performed may play a role in the IGF-1 response [22]. High-intensity interval training has been shown to promote IGF-1 increases in overweight women [29] and untrained men [30], concomitant with increasing insulin sensitivity, which plays a role in the indirect stimulation of IGF-1 production through increased growth hormone receptor sensitivity, leading to increased IGF-1 production [31]. Conversely, endurance training may produce a negative energy balance or prolonged metabolic stress in individuals, potentially leading to a decrease in IGF-1 production [32].

Studies investigating the effects of combined resistance and aerobic training on IGF-1 in healthy adults are limited and are fewer in women, particularly in pre-menopausal women. One study by Nindl and colleagues [33] found IGF-1 to remain stable in untrained, younger women (18–26 years) following eight weeks of combined exercise training. Therefore, the present study was designed to evaluate the effects of post-exercise protein supplementation with twelve weeks of combined resistance and interval training in healthy, pre-menopausal adult women on total IGF-1 concentration, as well as phenotypic and functional training adaptations. It was hypothesised that the combined effect of post-exercise protein supplementation and training would provide greater increases in total IGF-1 concentration, lean mass, and strength compared to the control group.

## 2. Materials and Methods

### 2.1. Study Design

A double-blind, randomised controlled trial was implemented to determine the effects of twelve weeks of combined upper-body resistance and interval training accompanied by a post-exercise protein supplement on outcome variables. The primary outcome measure was total IGF-1 concentration. Secondary outcome measures included dietary information, upper-body strength, and body composition. Prior to initiating this study, the protocol was reviewed and approved by the University of Otago Ethics Committee (H21/178) and registered through the Australian New Zealand Clinical Trials Registry (ACTRN12622000816752). The study was conducted in accordance with the Declaration of Helsinki, with written informed consent having been obtained from all participants. This study was conducted according to the Consolidated Standards of Reporting Trials (CONSORT) 2025 statement.

### 2.2. Study Procedures

#### 2.2.1. Screening

Healthy, pre-menopausal, untrained adult females (18–50 years of age), normally menstruating, without contraceptive use, were recruited for this study. Recruitment was conducted through online advertisements within the local Dunedin (New Zealand) community on social media, as well as university-listed emails and websites. A participant information sheet was provided upon initial communication, and prospective participants were sent a screening questionnaire to determine eligibility status based on inclusion criteria. Inclusion criteria were (a) not currently on a weight-loss programme; (b) not resistance-trained (no consistent resistance training within six months of commencing the study); (c) not competing in any competitive sport; (d) currently not injured and free from illness, including cardiovascular diseases, cancer, or any metabolic illnesses; (e) currently non-smoking/vaping; and (f) not taking any isotretinoin-based product. Weekly activity levels were also screened in the questionnaire, ensuring that participants were engaged in ≤1.5 h/week of vigorous physical activity. Additional criteria were (a) BMI > 18.5 < 30 (kg/m^2^) and (b) no contraindication to maximal exercise, as indicated by a physical activity readiness questionnaire (PAR-Q), which was measured upon the first in-person appointment.

Participants undertook twelve weeks of combined upper-body resistance and interval training and were randomly assigned to either the control group (CON) or moderate protein group (PRO). Participants were assigned a study number, and study numbers were randomly assigned to group one or two (through an online random number generator, https://www.randomizer.org/, accessed on 1 August 2022), which corresponded to either CON or PRO (see Intervention—Supplementation section for additional information). Training consisted of three weekly interval sessions and two resistance training sessions on separate days. While there were no rest days, the sessions were separated so that each participant had a minimum of 48 h between each resistance session. Additionally, if participants were unable to attend five sessions in a typical week, it was acceptable for participants to attend two daily sessions (one interval and one resistance session only), as long as there were eight or more hours separating each session in order to minimise potential training interference between exercise types [34].

Adherence to the training protocol was specified beforehand at ≥80% attendance over all sessions throughout the study. Training sessions were always supervised and were performed in small groups (five total), beginning at 7 a.m., 8 a.m., 12 p.m., 5 p.m., and 6 p.m. on every weekday, with a weekend session available upon request. Relevant measures were taken at baseline (T0) and during week four (T4), week eight (T8), and week twelve (T12).

#### 2.2.2. Intervention—Supplementation

Participants in CON were supplemented with Milo powder drinking chocolate (Nestlé, Auckland, New Zealand), whereas participants in PRO were provided with Nutratech chocolate-flavoured whey protein concentrate (Nutratech Ltd., Tauranga, New Zealand). Each serving was measured and placed into separate bags by the researcher (Table 1). Milo powder was selected due to its similar chocolate taste and its powder format. The amount of protein in the protein supplement was chosen, as this was an accurate representation of a “real-life” amount that would be in a normal protein shake. Supplementation was double-blinded, with each supplement being assigned a separate number (one or two) through an online random number generator, which corresponded to a separate room within the training facility. The placing and numbering of the supplements was carried out by a departmental staff member who was not associated with the study. After the first training session, participants were then instructed to fill up a protein shaker with water and then enter the room corresponding to their number to obtain, mix, and consume a single serving of their respective supplement. This process was then repeated after each session and after mid-week testing sessions, amounting to five servings of supplement per week. Researchers were informed of the supplement assignments at the end of the study after the analyses were completed.

#### 2.2.3. Intervention—Resistance Training Protocol

Participants completed two upper-body resistance training sessions per week, separated by 48 h between sessions to allow for sufficient recovery. Two participants trained at the same time to ensure appropriate safety and supervision from the researcher and to improve social motivation. Upper-body resistance training was split into three, 3-week training blocks, separated by the 3RM testing week, in which one session was replaced by testing (see Table 2). All exercises were completed with dumbbells. Chest press, single-arm dumbbell row, and shoulder press were the primary exercises completed every session. Two additional upper-body exercises as well as a short abdominal circuit were included on a rotational basis. Data for these additional exercises were not recorded, as their inclusion was to mimic a complete upper-body resistance session. Progression between testing was carried out according to the ‘rule of two’, meaning if a participant could complete two additional repetitions over the assigned number of repetitions for each set over two consecutive sessions, the load was increased [35]. Progression was reassessed and updated during testing sessions in weeks four and eight in accordance with the 3RM results and the prescribed intensity (see Table 2).

#### 2.2.4. Intervention—Interval Training Protocol

Interval training was completed three times weekly on a cycle ergometer. Progressive interval training was used, with sprint and rest intensities based on an RPE scale (1–10) [36]. Training was split into 4 × 3-week training blocks. The structure was the same for each block, apart from an added interval in each block, beginning at six intervals and finishing at nine intervals (see Table 3). The intensity during the sprint was considered ‘all-out’ (a ten on the RPE scale) with the rest phase being deemed as ‘active rest’ (a two on the RPE scale). Participants used a Polar H10 monitor chest strap (Polar Electro, Kempele, Finland) to record their heart rate during the session. These sessions were also logged into their respective Polar account to track the number of completed training sessions.

### 2.3. Measurements

#### Total IGF-1 Analysis (Primary Outcome)

Blood samples were collected at T0 and T12 from the antecubital vein into 4 mL lithium–heparin tubes, spun at 3000 RPM for ten minutes, and the plasma was removed and frozen at −80 °C until analyses. The samples were collected between 6–9 a.m., following an overnight fast (≥10 h), and participants were restricted from vigorous exercise and caffeine and alcohol consumption for ≥48 h prior to collection. The sample was taken during menses (on or between days 3–5), with the T0 sample taken as close as possible to the study commencing. The T12 samples were collected as close as possible to the end of the study. Participants were asked to inform the researcher when their menses began after week eight, since it was unlikely that the menses phase would align with the end of study. If menses fell before week ten, participants were asked to continue with the study protocol until the next cycle. If menses fell after week ten, the sample was taken in that phase of the menstrual cycle. Any time the sample was collected after T12, the study protocol was kept the same to avoid any potential changes in protein intake or detraining effects. If the sample collection date was during T12, the exercise testing was delayed until later in the week. Total IGF-1 concentration was measured in plasma by an accredited clinical laboratory (Canterbury Health Laboratories, Christchurch, New Zealand) by a chemiluminescence immunoassay via an automated ICA IDS-iSYS analyser (Immunodiagnostic Systems, Tyne and Wear, UK).

### 2.4. Dietary Food Records

Three-day food records were used to analyse participant nutrient intakes through an image-based dietary application (MealLogger, Version 4.7.4, Wellness Foundry, New York, NY, USA), at T0, T4, T8, and T12. Participants were asked to record two weekdays and one weekend day. Additionally, it was asked that all foods and liquids (except water) were recorded. If able, participants were to include nutritional information in the photo or caption to ensure as much accuracy as possible, as well as weighing of ingredients for meals. Participants were asked to maintain their regular diet throughout the study and not to begin taking any additional supplements outside of what they may have already been on. Food records were analysed through the FoodWorks 10 nutrition analysis software (Xyris Party Ltd., Brisbane, Australia).

### 2.5. Body Composition

Body mass and height were collected during each testing phase. Measurements for lean mass, fat mass, visceral adipose tissue, and bone mineral density were taken by dual-energy x-ray absorptiometry (DXA) (Lunar iDXA, GE HealthCare, Chicago, IL, USA) during T0 and T12 at the local hospital, and this was performed and analysed by a trained technician.

### 2.6. Strength Testing

During T0, T4, T8, and T12, participants underwent 3RM testing of three exercises: chest press, single-arm dumbbell row, and seated shoulder press. A short familiarisation session was conducted during the first testing session, prior to the secondary testing session where the 3RM testing took place. To begin the 3RM testing, participants warmed up at a self-selected, low-intensity weight and performed 8–10 repetitions of the respective exercise. After a 60–90 s rest, the following warm-up set consisted of another self-selected intensity but relatively heavier in load, aiming for five repetitions. After a three-minute rest, the third set was the beginning of the intended 3RM testing. If participants were successful in achieving three repetitions on this given weight—and deemed of good enough quality by the researcher—the weight was increased for the next set, following another three-minute rest. The three-repetition process was repeated until (a) the participant could not complete three repetitions or (b) the three repetitions were completed but the technique had decreased in quality to where potential injury risk was involved. The researcher aided in spotting the weights from a safety perspective and helped provide verbal motivation. This process was repeated for all three key exercises, with chest press always first in order, followed by the row, and finally the shoulder press. During T4 and T8, the 3RM testing was completed on the first resistance training session of the week. This allowed participants to be fully rested from the weekend (48 h) and ensured continuity of the training protocol.

After completion of the three exercises, the results (singular dumbbell weight) were entered into an online one-rep max (1RM) calculator (https://www.strengthlog.com/1rm-calculator/, accessed on 4 August 2022), which utilised Epley’s equation to determine the 1RM weight [37], which has been shown to be an accurate predictor of the 1RM weight [38]. Depending on the prescribed repetitions in the training protocol, a percentage of the 1RM weight was used to inform assignment of participants’ training loads (see Table 2).

### 2.7. Statistical Analyses

#### 2.7.1. A Priori Sample Size Calculation

Based upon prior work in previously active males [39] who participated in multimodal exercise over twelve weeks and found a 12% increase (PRE: 98.6 ± 78, POST: 110.5 ± 8.3 ng/mL) in total IGF-1 compared to a 10% decrease (PRE: 126.9 ± 13.8, POST: 116.9 ± 9.3 ng/mL) in the high and low protein diets, respectively, a sample size of eight per group was calculated to obtain adequately powered sample sizes (⍺ = 0.05, power = 0.08) [40]. Due to differences in population characteristics and training protocol between ours and those of Ives et al. [39], as well as multiple covariates potentially influencing the total IGF-1 concentration, we initially aimed to recruit 35–40 participants (allowing for 20% drop-out), with 15 participants per group.

#### 2.7.2. Analyses

Descriptive statistics are used in tables, where observed means and standard deviations are provided. Difference scores (means + standard deviations) are reported in written sections, and associated *p*-values are taken from the final models from linear mixed-effects model analyses. Linear mixed-effects model analyses were first run unadjusted, with group and time as fixed effects, participant as a random variable, and the interaction (group x time) included. When appropriate, covariates were included within the final adjusted models and are reported as such. Continuous predictor variables and covariates were mean-centred to reduce multicollinearity. The main effect of time, group, and interaction are reported in tables where appropriate. Analyses through linear mixed-effects models allowed for handling of missing data. Due to many potential covariates and the effect of the limited sample size on the outcome variables, the inclusion of covariates was limited to a maximum of three, and these were dependent on the best model fit according to the Akaike Information Criterion (AIC) value, which corresponded to the lowest AIC value. Covariates also warranted inclusion on a physiological basis in the final model(s). Dietary intakes were averaged across T4, T8, and T12 to generate one representative value to be used in the statistical models (the table in Section 3.5). Dietary intakes that were not averaged over the study duration can be found in Appendix A. Linear mixed-effects model analyses were also used to perform sensitivity analyses on all final adjusted models to compare participants who were adherent to the training protocol to participants who were not adherent. The mean difference (95% CI) between adherent and non-adherent participants is reported. Data analyses were conducted using IBM SPSS Statistics (Version 30.0). Standardised effect sizes (partial eta squared, ηp^2^) were obtained through the R Statistical Software, version 4.5.0 [41,42,43,44]. Effect sizes not reported in the written text are included in the Appendix A and labelled accordingly.

## 3. Results

### 3.1. Participant Screening

The Consolidated Standards of Reporting Trials (CONSORT) diagram for this study is presented in Figure 1. Participant recruitment started in June 2022, with the final testing sessions completed during April 2024. In total, 211 people reported interest in the study, with 78 (37%) responding to the pre-study questionnaire. Of these 78 individuals, 37 were included within the study, based on the inclusion/exclusion criteria outlined in the methods. A further six participants withdrew interest due to scheduling conflicts (*n* = 3), lack of compliance in pre-testing (*n* = 2), and pregnancy (*n* = 1). A total of 31 participants began the study, with 4 dropping out during the study, with reasons being external health issues (*n* = 1), lack of compliance with the supplement (*n* = 1), and other time commitments (*n* = 2). This resulted in 27 participants completing the study (CON = 12, PRO = 15). There were no adverse effects due to the training or protein supplementation.

### 3.2. Adherence

Training adherence was good in both the CON (90 ± 7%) and PRO (85 ± 9%) groups for the prescribed number of sessions. Attendance was similar in the resistance (91 ± 7%, 89 ± 7%, respectively) and cycle training sessions (89 ± 8%, 82 ± 13%, respectively). No significant differences in attendance were found between the groups (all *p* > 0.05).

### 3.3. Participant Characteristics

Participant characteristics are described in Table 4. Body mass did not change over time (*p* = 0.198, ηp^2^ = 0.02). Between groups, there was no difference in body mass over the twelve weeks (0.4 ± 1.5 kg, 0.1 ± 2.6 kg, CON, PRO, resp., interaction *p* = 0.975, ηp^2^ = 0.004). Similarly, there was no difference in BMI over time (*p* = 0.760, ηp^2^ = 0.009), with no difference between the groups over time (0.15 ± 0.49 kg/m^2^, 0.03 ± 0.80 kg/m^2^, CON, PRO, resp., interaction *p* = 0.961, ηp^2^ = 0.003).

### 3.4. IGF-1

No significant change in IGF-1 over the twelve weeks (*p* = 0.923, ηp^2^ = 0.0004) nor any difference between CON and PRO over time were observed (−6.0 ± 27.7 µg/L, −2.1 ± 27.8 µg/L, resp., interaction *p* = 0.920, ηp^2^ = 0.0005). In the final adjusted model, age and protein intake (g/kg lean mass) were included as covariates, and age was found to be a significant negative predictor of total IGF-1 concentration (−4.5 µg/L per unit increase in age (y), 95% CI; −6.5, −2.5, ηp^2^ = 0.51), but protein intake relative to lean mass was not (refer to Appendix A for complete statistical models). Energy, carbohydrate, and fat intakes were also included in prior models as covariates but were not significant. Other expressions of protein intake (absolute and relative to body mass) were also included in prior models yet were not significant.

### 3.5. Dietary Intake

Dietary intakes are reported in Table 5 (refer to Appendix A for dietary data every four weeks). Total energy intake did not significantly change over time (*p* = 0.439, ηp^2^ = 0.03), with no difference between CON and PRO over time (125 ± 1332 kJ, 336 ± 1510 kJ, resp., interaction *p* = 0.721, ηp^2^ = 0.006). Similarly, fat intake and carbohydrate intake did not significantly change over time (*p* = 0.570, ηp^2^ = 0.01 and *p* = 0.526, ηp^2^ = 0.02, resp.), and they were not different between CON and PRO over time (fat intake: −0.7 ± 23.0 g, −4.9 ± 24.0 g, resp., interaction *p* = 0.670, ηp^2^ = 0.008; carbohydrate intake: 13.8 ± 30.6 g, −1.5 ± 56.8 g, resp., interaction *p* = 0.432, ηp^2^ = 0.03).

Protein intake increased over time (*p* < 0.001, ηp^2^ = 0.42), with a greater increase in PRO (20.5 ± 13.9 g) than CON (5.0 ± 17.6 g) (interaction *p* = 0.025, ηp^2^ = 0.21). Similarly, protein intake relative to body mass also increased over time (*p* < 0.001, ηp^2^ = 0.44), with a greater increase in PRO (0.35 ± 0.25 g/kg BW) than CON (0.06 ± 0.23 g/kg BW) (interaction *p* = 0.011, ηp^2^ = 0.26). Protein intake expressed relative to lean mass also increased over time (*p* = 0.016, ηp^2^ = 0.33) and to a greater extent in PRO (0.48 ± 0.40 g/kg lean mass) than CON (0.05 ± 0.40 g/kg lean mass) (interaction *p* = 0.016, ηp^2^ = 0.24).

### 3.6. Body Composition

Body composition data are presented in Table 6. There were five total segmental measures included in both lean and fat mass outcomes, consisting of total, upper-body, trunk, arm, and leg segments. Total lean mass increased over twelve weeks (*p* = 0.009, ηp^2^ = 0.31), with no difference between CON and PRO over time (0.84 ± 0.80 kg, 0.56 ± 1.4 kg, CON, PRO, resp., interaction *p* = 0.750, ηp^2^ = 0.005). Improvements in upper-body lean mass were also found (*p* = 0.041, ηp^2^ = 0.2), but the groups were not different over time (0.46 ± 0.47 kg, 0.29 ± 0.94 kg, CON, PRO, resp., interaction *p* = 0.820, ηp^2^ = 0.003). Trunk lean mass did not significantly improve over twelve weeks (*p* = 0.339, ηp^2^ = 0.05), and these changes were similar between groups (0.23 ± 0.38 kg, 0.06 ± 0.94 kg, CON, PRO, resp., interaction *p* = 0.834, ηp^2^ = 0.002). Arm lean mass increased similarly in both CON and PRO (*p* < 0.001, ηp^2^ = 0.68), with similar increases observed between groups over the twelve weeks (0.23 ± 0.15 kg, 0.23 ± 0.16 kg, resp., interaction *p* = 0.913, ηp^2^ = 0.0006). Finally, leg lean mass significantly improved over the twelve weeks (*p* = 0.001, ηp^2^ = 0.42), yet the differences over time were similar between the groups (0.40 ± 0.50 kg, 0.28 ± 0.54 kg, CON, PRO, resp., interaction *p* = 0.659, ηp^2^ = 0.01). For all lean mass outcomes, IGF-1 and fat intake were included as covariates, yet neither were significant (refer to Appendix A). Energy, carbohydrate, and absolute protein intake were also included in prior models as covariates but were not significant.

No significant change in total fat mass was found over the twelve weeks (*p* = 0.269, ηp^2^ = 0.06), and no difference between the groups over time was observed (−0.46 ± 1.2 kg, −0.48 ± 1.9 kg, CON, PRO, resp., interaction *p* = 0.609, ηp^2^ = 0.01). Similarly, no changes over time were observed in upper-body fat mass (*p* = 0.374, ηp^2^ = 0.04), with no difference between the groups over time (−0.32 ± 0.81 kg, −0.32 ± 0.1.3 kg, CON, PRO, resp., interaction *p* = 0.825, ηp^2^ = 0.003). No change was observed for trunk fat mass over the twelve weeks (*p* = 0.325, ηp^2^ = 0.05), and the changes in both groups were similar over time (−0.31 ± 0.78 kg, −0.32 ± 1.3 kg, CON, PRO, resp., interaction *p* = 0.778, ηp^2^ = 0.004). Similarly, arm fat mass remained largely unchanged in both groups (*p* = 0.688, ηp^2^ = 0.008), with similar changes over time between the groups (−0.016 ± 0.18 kg, 0.010 ± 0.22 kg, CON, PRO, resp., interaction *p* = 0.98, ηp^2^ = 0.00003). Finally, there were no observed changes in leg fat mass over the twelve weeks (*p* = 0.240, ηp^2^ = 0.07), and the differences between the groups were similar over time (−0.13 ± 0.57 kg, −0.16 ± 0.63 kg, resp., interaction *p* = 0.384, ηp^2^ = 0.04). For each fat mass outcome, IGF-1 and protein were included as covariates but were not significant (refer to Appendix A). Energy, carbohydrate, and fat intake were also included in prior models as covariates but were not significant.

Additional measures from DXA included visceral adipose tissue, bone mineral density, and body fat percentage. Both visceral adipose tissue (−7.1 ± 61.4 g, −11.9 ± 114.6 g, CON, PRO, resp.) and bone mineral density (−0.01 ± 0.03 g/cm^2^, −0.01 ± 0.02. CON, PRO, resp.) did not change over the twelve weeks (*p* = 0.629, ηp^2^ = 0.01 and *p* = 0.809, ηp^2^ = 0.002, resp.), and they were not different between the groups over time (interaction *p* = 0.891, ηp^2^ = 0.0008 and *p* = 0.176, ηp^2^ = 0.07, resp.). Finally, body fat percentage decreased significantly over time (*p* = 0.016, ηp^2^ = 0.22), yet no differences between the groups over the twelve weeks were found (−1.8 ± 2.6%, −2.7 ± 5.7%, CON, PRO, resp., interaction *p* = 0.885, ηp^2^ = 0.0009).

### 3.7. Strength and Training Volume

Training volume and strength data are presented in Figure 2. For exact mean data, refer to Appendix A. Total training load for bench press, row, and shoulder press exercises were not different between the groups (all *p* > 0.05). Significant (*p* < 0.001, ηp^2^ = 0.87) increases over time were observed in both groups for 3RM chest press. CON increased by 7.0 ± 2.5 kg and PRO increased by 6.3 ± 1.7 kg, with no difference between the groups over time (interaction *p* = 0.781, ηp^2^ = 0.004). Similar increases over the twelve weeks (*p* < 0.001, ηp^2^ = 0.86) were also found in 3RM row, in which CON increased by 7.3 ± 2.0 kg, and PRO increased by 6.9 ± 2.1 kg, although no difference between the groups over time was found (interaction *p* = 0.493, ηp^2^ = 0.02). Lastly, 3RM shoulder press also significantly increased over time (*p* < 0.001, ηp^2^ = 0.83). CON increased by 7.0 ± 3.0 kg, with PRO increasing 6.2 ± 1.7 kg, although differences between the groups were similar over the twelve weeks (interaction *p* = 0.544, ηp^2^ = 0.02). Upper-body lean mass was a significant positive predictor for 3RM chest (0.52 kg, 95% CI; 0.11, 0.93, ηp^2^ = 0.22), 3RM row (0.75 kg, 95% CI; 0.35, 1.1, ηp^2^ = 0.37), and 3RM shoulder press (0.61 kg, 95% CI; 0.24, 0.97, ηp^2^ = 0.34). Absolute protein intake and IGF-1 were also included as covariates in the final adjusted models yet were not significant (Appendix A). Energy, carbohydrate, and fat intake were also included in prior models as covariates but were not significant.

### 3.8. Sensitivity Analyses

Sensitivity analyses were conducted on each final model for each outcome to compare adherence (≥44 total sessions, *n* = 22) versus non-adherence (*n* = 4) to the training protocol. No significant differences were found in any outcome variables apart from total IGF-1. While no differences between group or time were found for IGF-1 (*p* = 0.091, ηp^2^ = 0.14 and *p* = 0.195, ηp^2^ = 0.08, resp.), adherence to the programme was associated with reduced total IGF-1 concentration (−33.7 µg/L, 95% CI; −64.3, −3.1, interaction *p* = 0.033, ηp^2^ = 0.22). Refer to Appendix A for statistical models of sensitivity analyses.

## 4. Discussion

The purpose of the study was to examine the effects of twelve weeks of combined resistance and interval training with post-exercise protein supplementation on IGF-1, strength, and body composition in healthy, untrained pre-menopausal women. This study contributes to the limited knowledge surrounding the effects of post-exercise protein supplementation taken in conjunction with combined resistance and interval training in women.

### 4.1. IGF-1

It was hypothesised that total IGF-1 concentration would increase following twelve weeks of combined exercise training and would increase to a greater extent with a post-exercise protein supplement. The present findings do not support either hypothesis, in which insignificant decreases were observed in both the CON (~3%) and PRO (~1%) groups, with no difference between the groups over the twelve weeks. As expected, age was a significant negative predictor of IGF-1, as has been shown previously [45], with the decrease during menopause being particularly accentuated [46]. There was a large degree of individual variability, with changes ranging from a 28% decrease to a 41% increase. Across all participants, 56% had an increase in IGF-1, where 44% had a decrease in IGF-1.

To our knowledge, only two studies have examined the combined effects of protein supplementation and combined resistance and aerobic-based exercise on total IGF-1 concentration in healthy adults [24,39]. Ives et al. [39] investigated the impact of protein-paced meals (5–6 meals per day, each containing >0.25 g/kg protein, 2.0 g/kg per day) in men over a twelve week period of multimodal training (one resistance, endurance, pilates/yoga, and sprint interval session each week) on total IGF-1 concentration in resistance-trained men. In contrast to the present study, Ives et al. [39] observed a significant increase (~12%) in total IGF-1 in men following twelve weeks of high protein intakes combined with multimodal training. Another study by Ballard and colleagues [24], investigated the effects of six months of combined resistance and aerobic exercise training with a large protein bolus (42 g protein, 2.2 g/kg per day) taken twice daily compared to a carbohydrate supplement control (70 g carbohydrate) in untrained men and women. Ballard and colleagues [24] found significantly greater increases in IGF-1 in the protein group compared to the carbohydrate group. Both Ives et al. [39] and Ballard et al. [24]. had greater relative protein intakes compared to PRO in our study (1.5 g/kg per day), which may explain the differences seen in IGF-1. Elevated protein intakes have been associated with increased total IGF-1 concentration [20,47], and while not fully understood, it is thought that the increase in protein has a greater effect on GH stimulation, thus promoting hepatic production and release of IGF-1 [48] as well as increased amino acid availability within the bloodstream for IGF-1 synthesis [49]. While the evidence is limited in women in chronic training studies, Mallinson et al. [50] investigated the acute effects of post-exercise protein ingestion following repeated bouts of resistance exercise in trained, pre-menopausal women on IGF-1. The results from the study indicated that neither 15 g or 30 g protein supplementation was sufficient to elevate IGF-1 concentration at either 4, 8, or 24 h after exercise; however, a large protein bolus (60 g) significantly elevated IGF-1 concentrations 24 h following exercise [50]. Relative protein intake on the experimental day was expectedly high in the women assigned to the large protein bolus (3.2 g/kg). Interestingly, the relative protein intakes of the 15 g and 30 g protein group (1.8, 2.3 g/kg, resp.) were also much higher than intakes in our protein supplement group, yet they observed no change in IGF-1 in the 24 h post-exercise timeframe.

Nishida et al. [22] found that pre-training IGF-1 concentration was negatively correlated with change in IGF-1 following six weeks of low-intensity aerobic training in healthy men, and this was supported by another study in post-menopausal women undergoing resistance training [21]. While these associations have not been shown in healthy, pre-menopausal women, this may explain the lack of difference in total IGF-1 in our study compared to Ives et al. [39] and Ballard et al. [24], who reported substantially lower baseline concentrations of total IGF-1 (~40% and ~22%, respectively), thus resulting in a potentially greater ability to increase IGF-1.

Sensitivity analyses uncovered that participants who adhered to the training protocol experienced a significant decrease in IGF-1 in comparison to participants who were not adherent. It should be noted that the sample size was low in the non-adherent group (*n* = 4). The increased attendance of the training sessions may have resulted in larger energy expenditure in the participants, which, in turn, may have affected energy balance, which has been shown to directly influence total IGF-1 concentration [47,51]. An earlier study by Rarick et al. [52] theorised that energy flux (high levels of energy intake and expenditure under conditions of energy balance) may be responsible for the changes in total IGF-1 in response to short-term training (7 days), where a high energy flux decreased total IGF-1. While it is difficult to estimate energy flux in our study, based on estimations by Nindl et al. [33], who also found stable IGF-1 concentrations in young women in response to eight weeks of combined exercise training, we estimate that our participants may have been in low-to-moderate energy flux. Furthermore, energy balance was met on a group basis within the current study, and this is supported by the lack of change in body mass and the overall increases in lean mass. It should be noted that this area is largely unexplored and warrants further investigation to confirm these findings.

Finally, we found no effect of total IGF-1 on either body composition or muscle strength characterised by the lack of change in circulating IGF-1 in response to the training or dietary intervention within the study. Furthermore, analysis of total IGF-1 concentration indicates that overall production of IGF-1 was not altered, yet we cannot draw conclusions on the potential effects of training and dietary intervention on bioavailable IGF-1, as neither free IGF-1 nor binding proteins were measured. It should be noted that due to the hormonal and physiological differences in younger pre-menopausal women compared to older post-menopausal women, we have not compared our data with the extensive research on IGF-1 responses in post-menopausal women. Accordingly, we acknowledge that there is a clear gap within the literature regarding combined exercise training and protein supplementation in pre-menopausal women.

### 4.2. Diet

Post-exercise whey protein supplementation (24 g) significantly increased mean protein intake both absolute (20.5 g) and relative to body (0.35 g/kg) and lean mass (0.48 g/kg). Carbohydrate, fat, and total energy intakes did not change over the twelve weeks and were not different between the groups. These findings are in line with previous research, where Hida et al. [53] supplied female collegiate athletes with an egg white supplement (15 g protein) compared to an isoenergetic control over eight weeks and found that egg white supplementation increased average daily protein intake over eight weeks (0.15 g/kg) with no change in energy intake. However, these findings contrast with a cross-over study in resistance-trained males, in which short-term (two-weeks) high-protein (30 g protein, 1.72 g/kg) post-exercise supplementation was compared with a mixed meal supplement (10 g protein, 1.72 g/kg), and no observed increase in total protein intake was found, even though the proportion of protein increased (percentage of energy intake), due to a reduction in total energy intake [54]. These contrasting findings could be due to potential sex differences. One study investigated potential sex differences in overweight individuals who completed four exercise bouts with energy added to maintain energy balance and four bouts of exercise aimed to induce energy deficit [55]. The results indicated that in response to exercise, acylated ghrelin concentrations were greater in women (indicating stimulus to eat) following exercise, regardless of energy status, where no difference in either condition was reported in men [55]. Furthermore, Cornier et al. [56] suggested that women may exhibit different behavioural responses compared to men, as demonstrated by a higher sensitivity to food intake (through increased satiety ratings post-meal) in women, and that women were more likely to maintain an isocaloric intake during ad libitum feeding. This may explain the lack of change in energy intake seen in the present study and that of Hida et al. [53] and may be further explained through estrogen’s inhibitory actions in neuronal and hormonal regulators of appetite control [57]. Furthermore, although energy intake was unchanged in both groups in our study, protein intake has been shown to increase satiety and thermogenesis, which can further promote satiety and energy expenditure [58]. The proportion of protein in our participants’ diets may not have been high enough to induce the decreases in energy intake commonly seen in previous findings [59,60].

### 4.3. Body Composition

While investigations into protein supplementation following exercise have been extensively reviewed within the literature [61,62], research on the effects of combined resistance and interval training with post-exercise protein supplementation is limited. In the present study, increases in total lean mass were observed in both CON and PRO (2.0%, 1.8%, resp.). Although not significant, there was a favourable trend towards reductions in total fat mass in both CON and PRO (−1.2%, −2.1%, resp.). Contrary to our hypothesis, there was no additional effect of post-exercise protein supplementation on any body composition measure.

Body composition and muscular strength data from Ballard et al. [24] are reported by Ormsbee et al. [63]. Their sub-group analyses in previously untrained women found that differences between the control and supplement in protein intake (~1.0, 2.2 g/kg, resp.) were not correlated with changes in lean mass at three or six months, although greater protein did result in greater fat mass reduction at both three and six months. While our study did not show significant changes in fat mass over twelve weeks, our lean mass findings do support those of Ormsbee et al. [63]. However, sub-group analyses on men in their study revealed greater lean mass gains in the protein group at three months, yet changes in fat mass were similar in both groups [63]. Additionally, Josse et al. [64] found that twelve weeks of fat-free milk supplementation following resistance training in untrained women led to greater net gain in lean mass and greater reductions in body fat compared to an isoenergetic carbohydrate control whilst maintaining total body mass. Moreover, Ives et al. [39] found that in resistance-trained men, neither multimodal training nor higher protein intake affected lean mass, although training did reduce fat mass and percentage body fat. In contrast, in resistance-trained women using the same protocol, significant decreases in total fat mass and increases in lean mass were observed regardless of protein intake [65]. The differences in fat and lean mass changes amongst studies may be attributed to several factors, including energy balance, protein source, training load, training history, and hormonal status.

In the largest meta-analysis to date on protein supplementation with resistance training, Morton et al. [66] concluded that protein supplementation was more effective in resistance-trained individuals. It was speculated that, due to the attenuated muscle protein turnover and reduced potential for muscle growth in response to consistent long-term training [67], trained individuals may rely more on the effects of protein supplementation to attain an increase in muscle mass. Additionally, the authors indicated that resistance training alone is a much more potent stimulus in lean mass increases and may mask any effects of post-exercise protein supplementation, particularly in untrained individuals, which may be relevant to our findings. While the meta-analysis found no differences based on sex, it was acknowledged that there is much less work carried out in women than men. Based on this conclusion, it may be that twelve weeks of combined training with a post-exercise protein supplement in untrained women may not be long enough for significant changes in lean mass to occur from protein supplementation, hence the similarities seen regardless of protein intake. While we found increases in lean mass, we did not see any significant changes in fat mass outcomes, although our study had a high degree of variability within participants, highlighting the individual nature of adaptive responses and the multifaceted genetic and epigenetic effects on muscle growth [68].

### 4.4. Strength

Strength significantly improved over twelve weeks of progressive upper-body resistance training in previously untrained women, even when combined with aerobic exercise, as recommended in various weekly physical activity guidelines [69,70]. The mean increases in 3RM chest press (~48%), row (~41%), and shoulder press (~55%) demonstrate a strong response to the progressive resistance training protocol. Individually, there was a large degree of variability in chest press (~19–86%), row (~19–68%), and shoulder press (~24–113%).

Our strength gains are somewhat greater than those reported in a meta-analysis by Hagstrom et al. [71], where an increase (range 4–40%) in upper-body strength was observed in untrained women following whole-body resistance training programmes spanning an average of 15 weeks. Although Hagstrom and colleagues [71] found that training frequency ≥ 3 days/week induced greater effects on upper-body strength compared to less resistance training (1–2 days/week), our results indicate that significant gains in upper-body strength can be achieved with just two sessions per week. The fact that our training protocol used upper-body resistance exercises, rather than whole-body exercises, may explain the somewhat greater response than that reported in the meta-analysis [71].

In line with our results, Ormsbee et al. [63] found significant increases in 1RM bench press at both three (~12%) and six (~20%) months amongst untrained women, with no effects of protein supplementation, which has been supported in other previous work [64,72]. Interestingly, sub-group analyses on the men in their study found greater strength increases with greater protein intakes, although greater increases in protein are likely not directly associated with increases in muscle strength. While some evidence has shown differing rates of protein metabolism at rest and during exercise during the mid-luteal phase of the menstrual cycle [73,74], research is lacking in this field, and there are other important determinants that might play a role in the response to exercise training and protein supplementation. Interestingly, another meta-analysis investigating sex differences in resistance exercise on strength found a moderate effect size favouring women compared to men in upper-body strength gains in response to resistance training [75]. Furthermore, Arciero et al. [65], found that in resistance-trained women, higher protein intakes (2.0 g/kg per day) did not provide additional benefit to upper-body strength (1RM bench press); however, there was a significant effect of increased protein on upper-body and core muscular endurance (push-ups and sit-ups, respectively) as well as upper-body power (bench throw).

### 4.5. Limitations

There are several limitations within the present study. Firstly, the use of self-reported food records can lead to underreporting of food intakes [76], and the image-based approach to the food record yielded limited specific information about dietary composition and serving size. Additionally, while our supplementation protocol was controlled, we did not control for any meal timing during the study, and this may have had an overall impact of the efficacy of post-exercise protein supplementation, especially since our training times were prior to conventional mealtimes (breakfast, lunch, and dinner). It is likely that the participants ate a mixed meal (containing protein) shortly before or after training, making it difficult to discern any effect that the timing of protein after exercise may have had. Similarly, we have little insight into whether the participants trained fed or fasted, which may have produced different responses to the study. Another limitation was the analyses of only total IGF-1 concentration. While total IGF-1 is widely used to quantify and investigate IGF-1, there is no understanding of the distribution of IGF-1 in the body amongst its bioavailable or bound form. Given the effects of IGF binding proteins on the regulation and control of IGF-1 in the bloodstream [77], and that free IGF-1 is the bioactive form of IGF-1 in the body, future investigations may benefit from a broader approach to IGF-1 analyses. Additionally, by taking IGF-1 measurements at only two timepoints (T0 and T12), it is unknown if any changes in total IGF-1 concentration may have occurred prior to the end of the study during training adaptation. Our findings may not hold for younger (pre-pubescent) or, in particular, older (post-menopausal) women who are known to have reduced sex hormones and IGF-1 [46].

## 5. Conclusions

Women can significantly improve upper-body strength during concomitant resistance and interval training and can achieve favourable body compositional changes, following twelve weeks of training, irrespective of protein supplementation. Furthermore, these changes occurred in the absence of change in total IGF-1 concentration, suggesting that total IGF-1 production was unaltered. Future studies warrant continued investigations into the limited research space of protein supplementation combined with exercise protocols that are in line with globally recommended physical activity recommendations.

## Figures and Tables

**Figure 1 nutrients-17-02033-f001:**
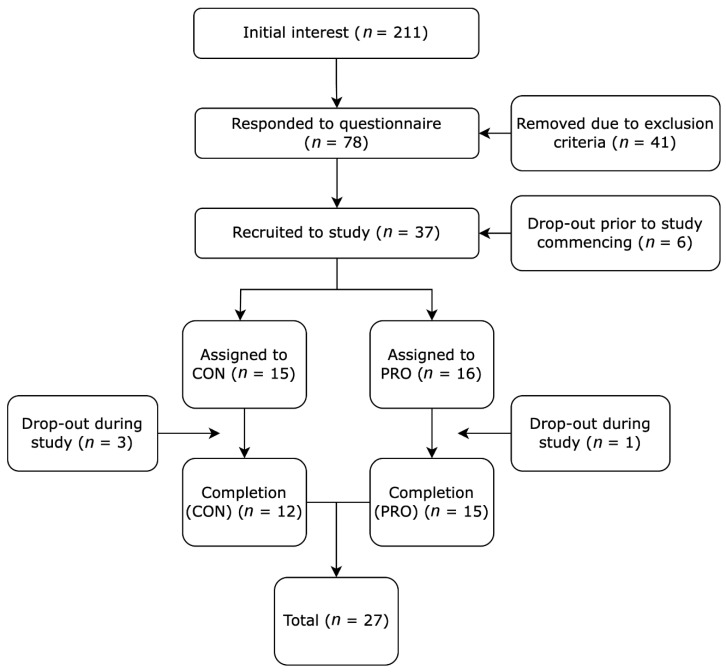
Flowchart outlining the progression of participants from initial interest through to the end of the study.

**Figure 2 nutrients-17-02033-f002:**
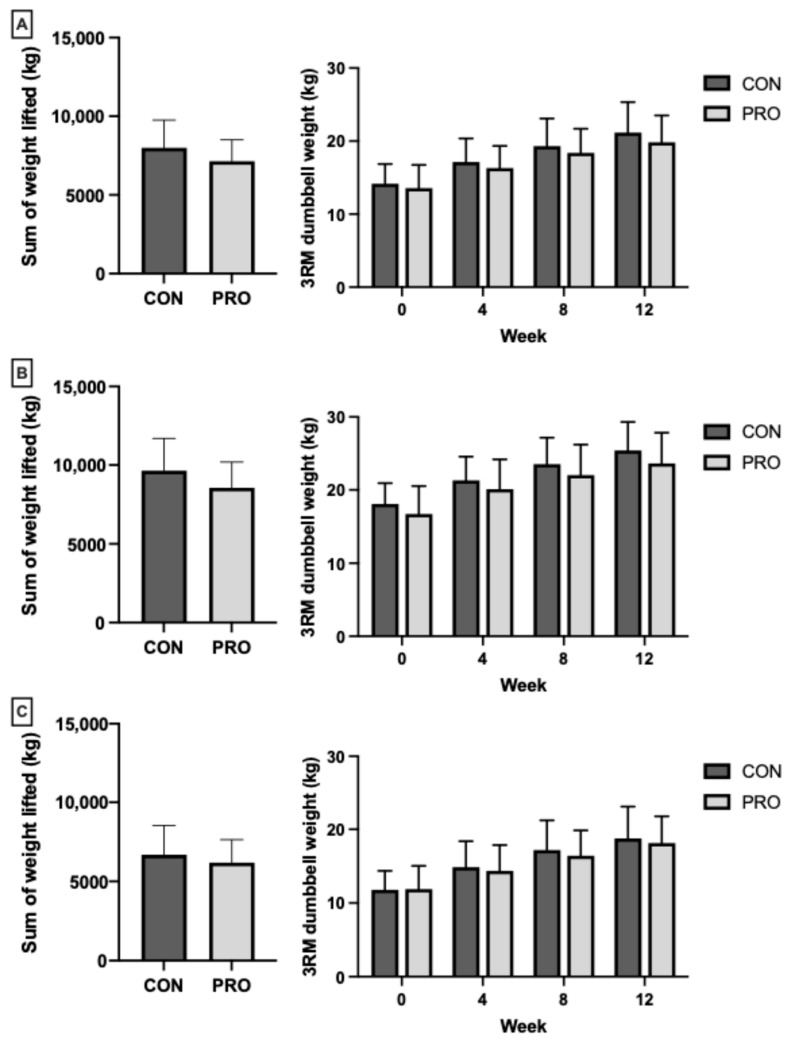
Total volume of weight lifted over twelve weeks and 3RM (mean ± SD) taken every four weeks for (**A**) chest press; (**B**) row; and (**C**) shoulder press between the control group (CON) and moderate protein group (PRO).

**Table 1 nutrients-17-02033-t001:** Serving size and composition of each supplement.

	CON	PRO
Serving size (g)	31.1	32.0
Energy (kJ)	510.4	510.4
Protein (g)	2.7	24
Carbohydrate (g)	21.7	1.7
Fat (g)	3.4	1.8

**Table 2 nutrients-17-02033-t002:** Resistance training protocol structure.

	Block 1	Testing	Block 2	Testing	Block 3
Sets	3	^X^	3	^X^	4
Repetitions	10	^X^	10	^X^	8
Load (%1RM)	75	^X^	75	^X^	80
Rest (s)	150	^X^	150	^X^	180

^X^ Testing week: one training session replaced by testing, the other at new training load.

**Table 3 nutrients-17-02033-t003:** Interval training protocol structure.

Block	Intervals *	Warm-Up	Work: Rest	Warm-Down	RPE Work: Rest
Week 1 & 2	6	5 min	60:90 s	5 min	10:2
Week 3	6	5 min	60:60 s	5 min	10:2

* Additional interval training block (up to nine).

**Table 4 nutrients-17-02033-t004:** Descriptive characteristics and total IGF-1 concentration between the control group (CON) and moderate protein group (PRO) measured at baseline (T0) and at the end of the study (T12).

	T0		T12		Mean Difference	Significance		
	CON (*n* = 12)	PRO (*n* = 15)	CON (*n* = 12)	PRO (*n* = 15)	T12	Group	Time	Interaction
Age (y)	32.8 (9.7)	34.2 (9.1)						
Height (cm)	167.3 (2.2)	163.3 (8.0)						
Body mass (kg)	75.3 (7.5)	64.2 (14.5)	75.7 (6.9)	64.4 (13.7)	−0.28 (−1.52, 0.97)	0.021	0.189	0.975
BMI (kg/m^2^)	27.1 (3.1)	24.0 (3.9)	27.2 (2.9)	24.1 (3.8)	−0.07 (−0.48, 0.33)	0.033	0.760	0.961
Total IGF-1 (µg/L)	203 (67.9)	178 (61.1)	197 (67.4)	176 (43.0)	1.38 (−26.6, 29.4)	0.280	0.923	0.920

Data are presented as observed means (SD) for respective timepoints. Mean difference (95% CI) between intervention relative to control group and significance data taken from linear mixed-effects model analyses. SD = standard deviation, *n* = number of participants, y = years, cm = centimetre, kg = kilogram, kg/m^2^ = kilogram/square metre, µg/L = microgram per litre.

**Table 5 nutrients-17-02033-t005:** Dietary information for the control group (CON) and moderate protein group (PRO) used in statistical analyses taken from baseline (T0) and during weeks four, eight, and twelve (T4, T8, and T12), where data from T4, T8, and T12 were combined to generate one timepoint (T4–T12).

	T0		T4–T12		Mean Difference	Significance		
	LP (*n* = 11)	HP (*n* = 13)	LP (*n* = 11)	HP (*n* = 13)		Group	Time	Interaction
Energy intake (kJ)	7742 (1361)	6868 (1743)	7867 (1092)	7205 (1133)	211 (−1001, 1425)	0.122	0.439	0.721
Protein intake (g)	79 (12)	70 (20)	84 (12)	91 (15)	16 (2, 29)	0.845	<0.001	0.025 *
Carbohydrate intake (g)	176 (65)	157 (55)	190 (43)	155 (56)	−15 (−55, 24)	0.200	0.526	0.432
Fat intake (g)	83 (19)	75 (30)	83 (23)	70 (16)	−4 (−24, 16)	0.209	0.570	0.670
Relative protein intake (g/kg BW)	1.1 (0.2)	1.2 (0.4)	1.1 (0.2)	1.5 (0.3)	0.27 (0.07, 0.47)	0.040	<0.001	0.011 *
Relative protein intake (g/kg lean)	1.8 (0.2)	1.9 (0.6)	1.9 (0.3)	2.3 (0.5)	0.43 (0.09, 0.77)	0.150	0.003	0.016 *

Data are presented as observed means (SD) for respective timepoints. Mean difference (95% CI) between intervention relative to control group and significance data taken from linear mixed-effects model analyses. * Indicates a significant (*p* < 0.05) group × time interaction effect. SD = standard deviation, *n* = number of participants, kJ = kilojoule, g = gram, g/kg BW = gram/kilogram of body weight, g/kg lean = gram/kilogram of lean mass.

**Table 6 nutrients-17-02033-t006:** Mean body composition data and analyses between the control group (CON) and moderate protein group (PRO) taken from DXA at baseline (T0) and at the end of the study (T12).

	T0		T12		Mean Difference	Significance		
	CON (*n* = 12)	PRO (*n* = 15)	CON (*n* = 11)	PRO (*n* = 15)		Group	Time	Interaction
Lean mass (kg):								
Total	43.3 (3.2)	40.2 (7.1)	44.3 (3.1)	40.7 (6.2)	0.46 (−0.54, 1.46)	0.028	0.009	0.75
Upper body	25.1 (1.9)	23.2 (4.0)	25.6 (2.0)	23.5 (3.6)	0.39 (−0.17), 0.96)	0.025	0.041	0.82
Trunk	20.4 (1.3)	19.1 (3.4)	20.7 (1.4)	19.1 (3.0)	0.15 (−0.71, 1.02)	0.041	0.339	0.834
Arm	4.7 (0.8)	4.1 (0.7)	4.9 (0.7)	4.4 (0.7)	−0.03 (−0.21, 0.14)	0.017	<0.001	0.913
Leg	15.1 (1.3)	14.0 (3.0)	15.6 (1.3)	14.3 (2.7)	0.07 (−0.44, 0.57)	0.048	0.001	0.659
Fat mass (kg):								
Total	28.8 (5.8)	21.5 (7.8)	28.4 (5.4)	21.0 (8.1)	−0.72 (−1.82, 0.39)	0.007	0.269	0.609
Upper body	17.2 (4.7)	16.9 (4.6)	12.2 (5.1)	11.9 (5.2)	−0.35 (−1.09, 0.40)	0.008	0.374	0.825
Trunk	13.9 (4.1)	9.8 (4.3)	13.6 (3.9)	9.5 (4.2)	−0.17 (−0.92, 0.57)	0.008	0.325	0.778
Arm	3.3 (0.7)	2.4 (0.9)	3.3 (0.7)	2.5 (1.0)	−0.03 (−0.28, 0.23)	0.019	0.688	0.98
Leg	10.8 (1.6)	8.4 (2.8)	10.7 (1.4)	8.3 (3.0)	−0.36 (−0.87, 0.14)	0.016	0.24	0.384
Body fat (%)	38.2 (4.5)	32.7 (5.8)	37.4 (4.2)	31.9 (6.4)	−0.09 (−1.36, 1.18)	0.012	0.016	0.885
VAT (g)	471 (317)	328 (297)	471 (307)	316 (308)	−5.3 (−83.8, 73.3)	0.22	0.629	0.891
Total BMD (g/cm^2^)	1.28 (0.01)	1.19 (0.07)	1.27 (0.10)	1.20 (0.07)	0.01 (−0.01, 0.03)	0.016	0.809	0.176

Data are presented as observed means (SD) for respective timepoints. Mean difference (95% CI) between intervention relative to control group and significance data taken from linear mixed-effects model analyses. SD = standard deviation, *n* = number of participants, kg = kilogram, g = gram, g/cm^2^ = gram/square centimetre.

## Data Availability

Dataset is available upon request from the authors. The data are not publicly available due to the personal nature of the data. Although no names included, with small sample size and body composition included, we feel that placing in a publicly available database could breach privacy.

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
