# Peer review of "Post-Exercise Whey Protein Supplementation: Effects on IGF-1, Strength, and Body Composition in Pre-Menopausal Women, a Randomised Controlled Trial"

_nutrients, 2025, doi:10.3390/nu17122033_

Round 1
Reviewer 1 Report
Comments and Suggestions for Authors
Dear Authors,
I am grateful for the opportunity to review your paper. There is much effort put into this investigation, so it is a pity that the final number of participants is not larger. At this stage, I consider your study a preliminary report (maybe it would be a good idea to include this in the title of your paper). If there were more subjects, you would obtain more significant results. However, I noticed that 211 individuals were initially interested in taking part in your study. Please enhance the resolution of Figure 2.
I hope you will enroll more participants in your next study!
Best regards,
The reviewer.
Author Response
“At this stage, I consider your study a preliminary report (maybe it would be a good idea to include this in the title of your paper).”
Thanks for your comments. We do not consider our results presented preliminary. These are the results of our two-year training study and effects of post-exercise protein supplementation on strength, body composition and IGF-1.
“Please enhance the resolution of Figure 2.”
Thank you for pointing this out, we have increased the resolution of Figure 2 as per the comment.
Reviewer 2 Report
Comments and Suggestions for Authors
This is a fairly well-conducted protein supplementation and exercise training intervention. Although this type of intervention is not very novel, the population (pre-menopausal women) is not assessed very frequently; therefore, the overall study is relatively novel.
I have the following suggestions for revision:
Title: I suggest changing Ifg-1 to IGF-1. I also suggest mentioning the type of protein used in the title (i.e., whey protein).
Some of the secondary outcomes mentioned in the clinical trials registry have not been presented in the manuscript. These include aerobic fitness, gut microbiota, and muscle soreness. If you have these data I suggest presenting them all in the same manuscript.
Page 5, statistics: Please either provide the means and SD or the effect size that went into the sample size calculation.
Please indicate whether there were any adverse effects with the training or protein supplementation. If there were no adverse effects, please include the statement “There were no adverse effects due to the training or protein supplementation.”
Figure 2: In the figure legend I suggest defining the abbreviations “HP” and “LP” used in the figure.
In the discussion you mention the meta-analysis by Morton et al. who suggested that protein supplementation had greater potential for benefit in individuals who are already trained. They also indicate that training by itself results in good increases in strength and lean tissue mass and this may mask benefits of protein supplementation especially in untrained populations. I suggest mentioning this point in your discussion section, as this may be the case in your study (i.e., the training by itself was very effective and may mask any small benefit of protein supplementation).
Author Response
Hi there - thank you for your in-depth review on our manuscript and some great suggestions. Please find the attached word document regarding our changes and answers to your comments.
Thank you.

Reviewer 3 Report
Comments and Suggestions for Authors
Introduction
It is well stated, a rationale is noted and objectives and hypotheses are stated, but it needs more explanation at the physiological level of the synergy of post-exercise protein intake.
Methods
In the section on participants, it is noted that the exclusion criteria did not include any type of supplementation that could alter the outcomes.
Explain how participants were randomised
Training consisted of three weekly interval sessions, and two resistance training sessions on separate days'. Please specify rest days between training sessions.
In the nutritional survey, is the application used validated?
Results
Please add the effect size in the tables.
Figure 3. Indicate the meaning of the acronyms LP and HP
Discussion
There is a lack of discussion of hormonal factors during the menopausal process that could affect each of the outcomes.
Author Response
Hi there - thank you for your in-depth review and suggestions on our manuscript, it is greatly appreciated. Please find attached a word document with our revised changes and answers to each of your feedback points.
Thanks!

Round 2
Reviewer 3 Report
Comments and Suggestions for Authors
The authors have responded appropriately to my comments.